# Myocarditis-like Episodes in Patients with Arrhythmogenic Cardiomyopathy: A Systematic Review on the So-Called Hot-Phase of the Disease

**DOI:** 10.3390/biom12091324

**Published:** 2022-09-19

**Authors:** Riccardo Bariani, Ilaria Rigato, Alberto Cipriani, Maria Bueno Marinas, Rudy Celeghin, Cristina Basso, Domenico Corrado, Kalliopi Pilichou, Barbara Bauce

**Affiliations:** 1Department of Cardiac, Thoracic, Vascular Sciences and Public Health, University of Padua, 35122 Padua, Italy; 2Azienda Ospedaliera di Padova, Via Giustiniani 2, 35128 Padova, Italy

**Keywords:** arrhythmogenic cardiomyopathy, myocarditis, troponin

## Abstract

Arrhythmogenic cardiomyopathy (ACM) is a genetically determined myocardial disease, characterized by myocytes necrosis with fibrofatty substitution and ventricular arrhythmias that can even lead to sudden cardiac death. The presence of inflammatory cell infiltrates in endomyocardial biopsies or in autoptic specimens of ACM patients has been reported, suggesting a possible role of inflammation in the pathophysiology of the disease. Furthermore, chest pain episodes accompanied by electrocardiographic changes and troponin release have been observed and defined as the “hot-phase” phenomenon. The aim of this critical systematic review was to assess the clinical features of ACM patients presenting with “hot-phase” episodes. According to PRISMA guidelines, a search was run in the PubMed, Scopus and Web of Science electronic databases using the following keywords: “arrhythmogenic cardiomyopathy”; “myocarditis” or “arrhythmogenic cardiomyopathy”; “troponin” or “arrhythmogenic cardiomyopathy”; and “hot-phase”. A total of 1433 titles were retrieved, of which 65 studies were potentially relevant to the topic. Through the application of inclusion and exclusion criteria, 9 papers reporting 103 ACM patients who had experienced hot-phase episodes were selected for this review. Age at time of episodes was available in 76% of cases, with the mean age reported being 26 years ± 14 years (min 2–max 71 years). Overall, 86% of patients showed left ventricular epicardial LGE. At the time of hot-phase episodes, 49% received a diagnosis of ACM (Arrhythmogenic left ventricular cardiomyopathy in the majority of cases), 19% of dilated cardiomyopathy and 26% of acute myocarditis. At the genetic study, *Desmoplakin (DSP)* was the more represented disease-gene (69%), followed by *Plakophillin-2* (9%) and *Desmoglein-2* (6%). In conclusion, ACM patients showing hot-phase episodes are usually young, and *DSP* is the most common disease gene, accounting for 69% of cases. Currently, the role of “hot-phase” episodes in disease progression and arrhythmic risk stratification remains to be clarified.

## 1. Introduction

Arrhythmogenic cardiomyopathy (ACM) is a genetically determined myocardial disease that is characterized by myocyte necrosis with fibrofatty substitution and the presence of ventricular arrhythmias, which can even lead to sudden cardiac death (SCD), especially in the young [1,2]. In approximately 50% of cases, a causative genetic variant can be found, which in most cases involves genes encoding for desmosomal proteins, although in recent years, some non-desmosomal disease genes have been identified [3].

After the first descriptions that considered the disease to be confined to the right ventricle (RV) [1,2], contrast-enhanced cardiac magnetic resonance (CE-CMR) studies demonstrated that the left ventricle (LV) is frequently involved [4]. Moreover, in 2008, Sen-Chowdhry et al. [5] described a clinical entity named “Arrhythmogenic left ventricular cardiomyopathy” (ALVC), characterized by a predominant LV involvement, with no or minor RV abnormalities. Thus, phenotypic expression of ACM has then been recognized as being wider than previously thought, including the “classical” right dominant form with exclusive RV involvement (Arrhythmogenic right ventricular cardiomyopathy: ARVC), the above-cited ALVC form and the biventricular form (BIV), diagnosed when both ventricles resulted as being involved in the disease [6].

Since the first disease descriptions, inflammatory cell infiltrates have been reported in endomyocardial biopsies or an autoptic specimens of ACM patients [1,2]. Furthermore, chest pain episodes accompanied by electrocardiographic changes and troponin release in keeping with acute myocarditis have been described and defined as “hot-phase” [1,2,7]. These episodes can in some cases be the initial presentation of ACM, and enter into differential diagnosis with acute myocarditis [8]. Currently, their role in disease progression, as well as in arrhythmic risk stratification, remains to be clarified

The aim of this critical systematic review of the literature was to assess the clinical features in ACM patients presenting with “hot-phase” episodes.

## 2. Materials and Methods

### 2.1. Study Plan

This study was conducted according to PRISMA guidelines (http://www.prisma-statement.org/) (Accessed on 2 July 2022). A search was run in the PubMed, Scopus and Web of Science electronic databases for clinical studies that investigated ACM patients. We collected published research using the following search items: “arrhythmogenic cardiomyopathy”; “myocarditis” or “arrhythmogenic cardiomyopathy”; “troponin” or “arrhythmogenic cardiomyopathy”; and “hot-phase”. MeSH terms and keywords were combined accordingly on the aforementioned databases. The reference lists of all the included articles were accurately screened in order to identify other pertinent studies. The ‘‘Related Articles’’ option on the PubMed homepage was also considered. No restriction about publication date was applied. Titles and abstracts of articles available in the English language were also evaluated. The full texts of the publications identified were screened for original data, and the references in the articles retrieved were checked manually for other relevant studies. The literature search has been updated to 2 July 2022.

### 2.2. Inclusion and Exclusion Criteria

Studies were included when the following general criteria were met: (1) articles were original reports; (2) reports were published in the English language; (3) studies only included patients who finally received the diagnosis of ACM (ARVC, BIV or ALVC forms) and who showed chest pain and myocardial release, and/or whose electrocardiographic (ECG) data changed. Multiple reports from the same data have been identified, as duplicate publications can introduce substantial biases if studies are inadvertently included more than once in an analysis [9]. Editorials, reviews and case reports were excluded.

### 2.3. Data Extraction

Two of the authors (B.B. and R.B.) extracted the data from the selected articles. Disagreements were dealt by discussion among the team members. Details of the search process and study selection are shown in Figure 1. We also reviewed the references of the included studies and systematic reviews to identify additional studies that were not captured by our database searches. Information extracted from the studies included the title, name of the first author, year of publication, country of study population and qualitative description of target population. Each included study was analyzed to extract all available data and ensure the eligibility of every single patient. For our review, we considered the following patients’ data: age at the episode of hot-phase; sex; family history of SCD; ACM/ALVC; myocarditis; dilated cardiomyopathy (DCM); presence of genetic variants; chest pain episodes; myocardial enzyme release (both troponin I, T or creatine phosphokinase-MB [CK-MB]); ECG abnormalities during hot-phase episodes (ST segment abnormalities, low LQRS voltages (LQRSv) or negative T waves in precordial or limb leads); ECG at rest; and major arrhythmic events (SCD, ventricular fibrillation [VF], sustained ventricular tachycardia [sVT], heart failure (HF), LV/RV dilation/dysfunction, subepicardial or ring-like late gadolinium enhancement [LGE] at CE-CMR and ICD implantation).

## 3. Results

The studies included in the systematic review are summarized in Table 1.

### 3.1. Study Retrieval

A total of 1433 titles were retrieved (414 from PubMed, 602 from Scopus and 417 from Web of Science). After removing duplicates, a total of 571 titles were screened, allowing us to identify 65 studies potentially relevant to the topic. The full-text screening of these articles led to the exclusion of 30 studies due to their non-compliance with the inclusion/exclusion criteria. Of the remaining 35 articles, 23 were excluded as consisting of case reports, and 3 due to the presence of duplicate data in other selected papers of the same research group [6,7,17]. Overall, 9 papers were finally considered for this review (Table 1) [5,8,10,11,12,13,14,15,16]. A PRISMA flow diagram depicts the flow of information through the different literature review phases (Figure 1).

### 3.2. Clinical Feature of Patient with Hot-Phase Episodes

Overall, 9 studies reporting 103 ACM patients with a history of hot-phase episodes were finally analyzed. The definition of “hot-phase episode” provided in each study is reported in Table 2. The mean observed incidence of hot-phase was 12 ± 7% (min 2%–max 22%). In 56 cases (54%), sex was reported, showing a mild prevalence of males (*n* = 33, 59%) (Table 1). Age at time of episodes was available in 78 patients (76%), with the mean age reported being 26 years ± 14 years (min 2–max 71 years). In 22 patients (21%) chest pain with myocardial enzyme release occurred more than once (max 6 reported episodes). Basal ECGs were available in 43 patients (42%) and were within normal limits in 14 (33%), while 8 patients (19%) showed negative T waves in V1-V3, 12 (28%) negative T waves in V5-V6, 7 (16%) negative T waves in inferior leads and 2 (5%) LQRSv. Moreover, ECGs changes during symptoms were reported in 15 patients (35%), mostly consisting of ST segment abnormalities.

### 3.3. Diagnosis of a Cardiac Disease in Patients with Hot-Phase Episodes

Data on ventricular dimension and function were available for 58 patients (56%). A dilated LV was reported in 21 patients (36%), and LV showed a reduced systolic function in 22 (38%). Analysis of RV parameters showed a dilated RV in 19 patients (33%) and a reduced RV function in 13 (22%).

Overall, in 79 patients, data on late gadolinium enhancement (LGE) were reported; 68 (86%) showed LV epicardial LGE, while RV LGE was reported in 56 patients, and its presence was detected in 9 cases (16%). Figure 2 shows CE-CMR images obtained during two different episodes of chest pain in a patient affected by ACM.

At the time of hot-phase episodes, 91 patients (88%) received a diagnosis: ACM (*n* = 44, 48%), DCM (*n* = 17, 19%), acute myocarditis (*n* = 24, 26%), acute myocardial infarction in the presence of normal coronary arteries (*n* = 6, 7%). Of the ACM group, the majority of patients (61%) showed an ALVC form. ACM diagnosis during or at the end of follow-up was reported in 59 patients (57%), of whom 40 (68%) were diagnosed with ALVC and 19 (32%) with right dominant or biventricular ACM forms.

### 3.4. Arrhythmic Symptoms and HF in Patients with Hot-Phase Episodes

With regards to the degree of electrical instability during hot-phases, data were available for 56 patients (54%). Overall, 5 patients (9%) showed SCD, and 5 (9%) sVT. Considering all life history, data on ventricular arrhythmias or arrhythmic symptoms were available for 58 patients (56%); of these, 11 (20%) had SCD episodes and 10 (17%) sVT. Finally, 4 patients (7% of available data) experienced HF.

### 3.5. Genetic Background in Patient with Hot-Phase Episodes

A genetic study was available for 94 patients (91%), and *DSP* was the more represented disease-gene (*n* = 65 patients, 69%), followed by *PKP2* (*n* = 8, 9%) and *DSG2* (*n* = 6, 6%). Three patients were found to carry multiple desmosomal mutations. In 9 cases (10%), the genetic test resulted as being negative. In only 49 out of 85 patients (58%) with a positive genetic test, detailed information regarding the genetic variants was available (Table 3).

## 4. Discussion

### 4.1. Hot-Phase in ACM Patients: An Historical Perspective

The presence of inflammatory infiltrates in myocardial samples of patients affected by ACM has been reported since the first disease descriptions. In 1990, Hisaoka et al. described two patients with clinical diagnosis of ACM, whose autopsies were in keeping with chronic myocarditis [18]. In the same year, Sabel et al. reported a 12-year-old girl with a familial form of ACM in whom ECGs and serum enzymes indicated the development of LV infarction [19]. The authors suggested that myocarditis could be a precipitating factor in patients showing an ACM form. A few years later, Hoffmann et al. described a 47-year-old man who was diagnosed with acute myocarditis, and later had an aborted SCD. The patient was later diagnosed with ACM, which was considered to be mimicked by chronic myocarditis [20].

In 1996 Basso et al. performed a pathological study in 30 hearts with ACM, and found scattered foci of lymphocytes with myocardial death in 67% of cases [21]. One of the interpretations of these findings was that the disappearance of the RV myocardium, which constitutes the diagnostic hallmark of ACM, could be interpreted as the consequence of an inflammatory necrotic injury followed by fibrofatty repair, and thus an infectious and/or immune myocardial reaction might intervene in the etiology and pathogenesis of the disease [21]. After the identification of defects in desmosomes components as the genetic basis of the disease, it was hypothesized that the gene defect could predispose to myocyte detachment and death, and that a significant myocyte loss may be accompanied by an inflammatory response [22]. In 2007, these periodic exacerbations of an otherwise-quiescent disease, clinically characterized by chest pain and myocardial enzyme release, were defined for the first time as “hot-phases” [22].

### 4.2. Hot-Phase in ACM Patients: General Considerations on Available Clinical Studies

While in the last few years the possible role of inflammation in ACM onset and progression led to a significant number of publications, few detailed clinical studies on a large cohort of patients with “hot-phases” can be found [23,24,25,26]. We were able to select only 9 studies describing cohorts of patients affected with ACM who showed myocarditis-like episodes [5,8,10,11,12,13,14,15,16]. This is probably due to the low prevalence of these episodes in the overall ACM population, and on the other hand, to the problem of differential diagnosis with acute myocarditis. It is noteworthy that diagnostic assessment in patients with chest pain and troponin release changed importantly with the wider use of CE-CMR, and with the identification of ALVC forms in 2008 [5].

### 4.3. Hot-Phase Episodes in ACM Patients: Clinical Features

Patients reported with hot-phase episodes were usually young, with a mean age of 26 ± 14 years; the pediatric population was highly represented [8,11,12]. Martins et al. reported six cases of ACM in children (mean age 9 years, min 2–max 15 years) carrying desmosomal gene mutations with evidence of myocardial inflammation at CE-CMR. Interestingly, hot-phase episodes were likely exercise-induced in 50% of cases [11]. De Witt et al. reported 32 children and adolescents with ACM diagnosis, and found troponin release and CE-CMR features compatible with myocardial inflammation in 6 cases (min age 6 years) [12]. Finally, Bariani et al. evaluated 23 patients with hot-phase episodes, of whom 12 were younger than 18 years (min 10 years) [8]. We could speculate that inclusion of studies that evaluate only pediatric patients could induce bias; however, the two selected cohorts of pediatric patients represent a small number of the total amount of patients analyzed (11/103, 11%) [11,12]. Furthermore, analysis of the 92 patients described in the remaining 7 studies showed that in 16 cases (40%), the age at the time of hot-phase was below the age of 18.

In addition, during chest-pain episodes, ECG can show ST segment abnormalities that could lead to the diagnosis of acute coronary syndrome or acute myocarditis (Figure 3). The above data suggest that in young patients presenting with myocarditis-like syndrome, a careful familial anamnesis searching for family history of SCD, myocarditis, or cardiomyopathy is mandatory. In this setting, a complete family screening, with a genetic study, should be indicated.

### 4.4. Hot-Phase and ALVC Forms

In our review, about two-thirds of patients who experienced hot-phase episodes were finally diagnosed with ALVC, often after a previous diagnosis of myocarditis. This is not surprising, as a clinical overlap exists between ALVC and myocarditis, considering that chest pain with ECG abnormalities, troponin release and LV LGE are present in both conditions. At the same time, a diagnosis of ARVC and BIV forms was achieved in one-fourth of patients. Differently from the other studies, Bariani et al., among 23 patients reporting hot-phase episodes, found 82% of ARVC and BIV forms, while ALVC forms accounted for 17% of cases [8]. Similarly, De Witt et al. [12] identified 6 pediatric patients with hot-phases and diagnosed an ARVC/BIV form in 4 of them. This difference from the other studies could be explained by the different selection of patients. Nonetheless, these data demonstrate that hot-phase episodes can characterize the whole ACM clinical spectrum, even if ALVC forms are more commonly found.

### 4.5. Hot-Phase in ACM Patients: Genetic Background

In the last few years, genetic studies allowed the identification of several ACM disease genes [27]. With the development of next-generation sequencing, molecular genetic testing for multiple genes using a multigene panel has become the standard of practice for cardiovascular genetic medicine, allowing a cascade screening of family members [28]. This led to increasing knowledge on genotype-phenotype correlation in ACM and genetic variants of genes encoding for desmosomal proteins, Phospholamban and Filamin-C, have been proven to be significantly present in ALVC forms [29]. Considering the presence of a clinical overlap between chronic myocarditis and ALVC, genetic testing has been proposed as an effective tool facilitating the differential diagnosis [30]. In our review, we found that in ACM patients who showed hot-phase episodes, *DSP* was the most common disease gene, accounting for 69% of cases. It should be emphasized that in two of our selected studies, one of the inclusion criteria was the presence of pathogenic variant of the *DSP* gene, and this may have resulted in an overestimation of the incidence of hot-phase episodes in this genotype [14,15]. Despite this, even excluding these two studies, *DSP* remains the gene most frequently found associated with hot-phase [8,10,11,12,13,14,16]. On the other hand, genetic variants of *PKP2* and *DSG2* were detected in 17% of cases. These data confirm the key role of a genetic test in diagnostic work-up of cardiomyopathies, even if it is important to consider that gene-elusive cases are not rare, and consequently, in these patients, further investigations must be carried out to rule out phenocopies [31].

### 4.6. Arrhythmic Burden in ACM Patients with Hot-Phase of the Disease

We believe this to be one of the most important issues when analyzing patients with hot-phases, given that this phenomenon can sometimes be long-lasting, and that recurrences are common.

Unfortunately, data on degree of electrical instability during the hot-phases are limited, and studies are difficult to compare due to the different selection of patients. Overall, 9% of patients showed SCD during the acute phase of the disease, and 9% had sVT episodes, even if data were available only in 54% of our cohort. The only study that demonstrated a possible prognostic role of hot-phase episodes is that of Wang et al. [15], which identified 91 individuals (34% male, median age 27.5) carrying a pathogenic or likely pathogenic *DSP* variant, and found that proband status and myocardial injury were prognostic for HF in univariate analysis, while in multivariate analysis these two variables did not reach statistical significance. Thus, data seem to indicate the presence of a significant degree of electrical instability during hot-phase episodes, requiring a continuous ECG monitoring in a hospital until the end of the acute episode. However, data on arrhythmic risk during follow-up are lacking. Future studies on a sufficiently large cohort of ACM patients are needed in order to reach definitive conclusions on arrhythmic risk and possible indication for ICD implantation in patients with hot-phase episodes.

### 4.7. Role of Inflammation in the Pathogenesis of ACM

Since the first reports of the disease, inflammatory cell infiltrates have been described in patients affected with ACM at postmortem and at endomyocardial biopsies, with the highest prevalence in people with more diffuse diseases [32]. Several studies suggested that inflammation, either reactive to internal influences or triggered by exogenous factors, has a role in the pathogenesis of ACM. In addition, most of the evidence on the role of inflammation and autoimmunity in this disease derives from studies on desmosomal forms of disease, while non-desmosomal, as well gene-elusive forms, require further studies to assess the role of inflammation and of possible modifier factors [23,24,25,26,33]. Studies on murine models supported the evidence that ACM inflammation can precede the onset of overt histological and electrical abnormalities [34]. However, the mechanistic link between cell death and inflammation remains to be fully explained [35]. Cardiomyocyte death could be a primary event due to genetically determined desmosomal disruption, enhanced by a secondary immune reaction, leading to amplified cell death [23,24,25,26]. In addition, circulating anti-DSG2 autoantibodies, anti-heart autoantibodies (AHAs) and anti-intercalated disk autoantibodies (AIDAs) were identified in patients with ACM, suggesting that autoimmune response against intercalated disk components and myosin could play a role in the pathogenesis of the disease [36,37]. In humans, the level of anti-DSG2 antibodies has been found to correlate with the arrhythmic burden, AHAs to be associated with lower LV systolic function and ICD indication, and AIDAs with lower biventricular systolic function [37]. Remarkably, anti-DSG2 antibodies have been detected in ACM regardless of the presence or type of the underlying pathogenic variant, thus suggesting that a final common pathway can underlie gene-elusive ACM patients [23,36].

It is noteworthy that, regardless of evidence of inflammation in ACM pathophysiology, studies of inflammatory pathways in this disease are not fully explained, and moreover, they have not so far been translated to human diseases [36]. Current therapeutic strategies are limited to the prevention of SCD through lifestyle changes, use of antiarrhythmic drugs, ICD placement, catheter ablation and treatment of HF symptoms. In overt forms with refractory HF, heart transplants represent the ultimate therapeutic strategy [24]. Thus, understanding the role of inflammation and autoimmunity could introduce new targets, potentially leading to new therapeutic strategies [23,24,38].

## Figures and Tables

**Figure 1 biomolecules-12-01324-f001:**
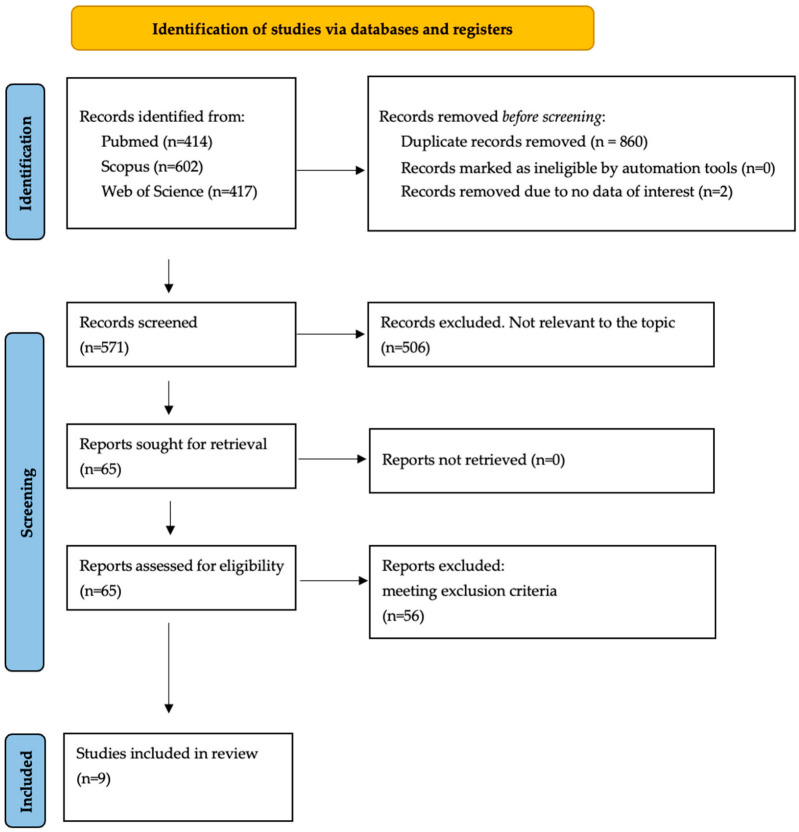
PRISMA flow diagram summarizing the literature review and inclusion/exclusion process.

**Figure 2 biomolecules-12-01324-f002:**
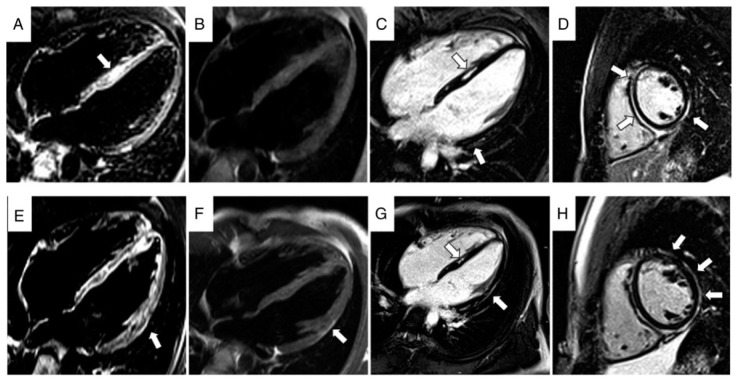
CE-CMR images obtained during two different episodes of chest pain, with 3 years interval [first episode (**A**–**D**) and last episode (**E**–**H**)]. In (**A**,**E**), note the presence of oedema (arrow) in the septum (**A**) and in the lateral wall of LV (**E**). In (**B**,**F**) turbo spin echo sequences demonstrate the evolution with appearance of a fat subepicardial stria in lateral wall of LV (arrow). Inversion recovery sequences obtained after administration of contrast agent (**C**,**D**,**G**,**H**): note the extension of fibrosis from septum and inferior wall to lateral wall during years (arrows). LV, left ventricle. Reproduced with permission from Bariani et al., Europace, 2021 [8].

**Figure 3 biomolecules-12-01324-f003:**
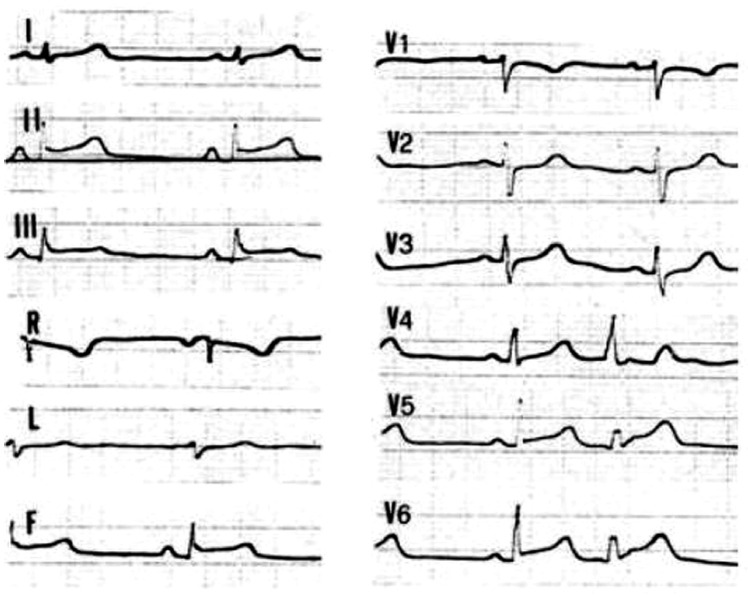
ECG of a 17-year-old female patient during a hot-phase episode. Note the presence of ST segment elevation in inferior and lateral leads. The patient belongs to an ACM family and carries a DSP genetic variant. Reproduced with permission from Bauce et al., European Heart Journal 2005 [7].

**Table 1 biomolecules-12-01324-t001:** Studies included in the systematic review.

Reference	Aim and Design of the Study	Study Population	Main Results	Patients with Hot-Phase	Genetic Variants Detected in Patients with Hot-Phase	Conclusions
Sen-Chowdhry, JACC 2008 [5]	Aim: to investigate the clinical-genetic profile of ALVC patients	42 patients (22 M) with ALVC	- Patients showed arrhythmia or chest pain, but not HF - Desmosomal mutations identified in 45% of patients - In 50% previously diagnosed with viral myocarditis or DCM	11 patients	*DSP*, *PKP2*, *DSG2*	-ACM is distinguished from DCM by a propensity towards arrhythmias exceeding the degree of ventricular dysfunction.
Lopez Ayala, Hearth Rhythm 2015 [10]	Aim: to evaluate the genetic basis of myocarditis in ACM and investigate the association with a poorer prognosis and a higher risk of ventricular arrhythmias	131 patients - 84 ACM (62% M) - 47 ALVC (47% M)	- Hot-phase as first clinical presentation in 6 out of 7 cases - In 2 patients, hot-phase episode preceded a worsening of LV systolic function, and in an additional 2 patients an episode of ventricular tachycardia	7 patients (3 M, 4 F), mean age at symptom 30 years (min 14-max 45 years) 3 ARVC, 2 ALVC, 2 NR	*DSP*, LIM domain binding protein	-Acute myocarditis reflects an active phase of inflammation in ACM, leading to changes in the phenotype and abrupt complication of the disease. -An active phase of inflammation should be suspected in the presence of myocarditis associated with a family history of ACM.
Martins, Int J Cardiol 2018 [11]	Aim: to study the relationship between myocardial inflammation detected at CMR and ACM in a pediatric population	ACM patients < 18 years with clinical suspicion of myocarditis who had genetic testing for inherited cardiomyopathies	Six ACM patients experiencing myocarditis-like episodes with chest pain and troponin elevation. - Hot-phase episodes were likely exercise-induced in 50% of cases	6 patients (5 M, 1 F), mean age at symptoms 9 years (min 2, max 15 years) 1 ARVC, 3 BIV, 2 ALVC	*DSP*, *PKP2*, *DSG2*	- ACM can present as recurrent myocarditis-like episodes with CMR evidence of myocardial inflammation despite absent infectious trigger in children.
De Witt, JACC 2020 [12]	Aim: to describe the diverse phenotype, genotype, and outcomes in pediatric and adolescent patients affected with ACM	ACM patients < 21 years, divided into three groups (ARVC, ALVC, BIV forms)	- 32 patients (mean age 15.1 ± 3,8 years), 22 probands (16 ARVC, 9 BIV, 7 ALVC)	6 patients, min age 6 max 21 years. 1 ARVC, 3 BIV, 2 ALVC	*DSP*, *DES*, *PKP2*	-ACM in the young has highly varied phenotypic expression incorporating life-threatening arrhythmias, HF and hot-phases.
Piriou, ESC Heart Failure 2020 [13]	Aim: to assess the risk of patients with acute myocarditis of carrying an associated genetic variant involved in familial cardiomyopathies	Families with at least one individual with a documented episode of acute myocarditis and at least one individual affected with a cardiomyopathy or with a history of SD	- Six families (33 subjects) were identified - In the 5 families with a *DSP* variant, genetic testing was triggered by the association of an acute myocarditis with a single case of apparent DCM or SCD - Among 28 *DSP* variant carriers 39% had ALVC - Family history of SCD was frequent	6 patients (3 M, 3 F), mean age at symptoms 20 years (min 9, max 41 years) 1 BIV, 5 ALVC	*DSP*, *DSG2*	-Comprehensive familial screening, including genetic testing in the case of acute myocarditis associated with a family history of cardiomyopathy or SCD, revealed unknown misdiagnosed ALVC patients. -Genetic testing should be advised in patients who experience acute myocarditis and have a family history of cardiomyopathy or SCD.
Smith, Circulation 2020 [14]	Aim: to systematically analyze the clinical spectrum of *DSP* cardiomyopathy	107 patients with *DSP* mutations and 81 with *PKP2* mutations identified at 6 tertiary referral centers of DCM and ACM	- ALVC forms were exclusively present among patients with *DSP* mutation carriers -LV EF <55% strongly associated with severe ventricular arrhythmias in DSP patients -RVEF < 45% associated with severe arrhythmias in PKP2 group	16 out of 107 *DSP* mutation carriers (15%) with troponin release 1 BIV, 15 ALVC	*DSP*	-*DSP* cardiomyopathy is a distinct form of ACM cardiomyopathy, characterized by episodic myocardial injury, left ventricular fibrosis and a high incidence of ventricular arrhythmias.
Bariani, Europace 2021 [8]	-Aim: to evaluate the clinical features of patients affected by ACM presenting with chest pain and myocardial enzyme release in the setting of normal coronary arteries (‘hot-phase’)	-ACM patients presenting with chest pain and/or myocardial necrosis markers elevation in the setting of normal coronary arteries	-Among 530 patients fulfilling ARVC TFC, 23 (5%) experienced hot-phase episodes - Genetic testing was positive in 77% of cases and pathogenic -*DSP* was the most frequent involved gene - No patient complained of sustained ventricular arrhythmia or died suddenly during the hot-phase	23 patients (12 M, mean age at symptoms 24 years, min 10–max 71 years) 5 ARVC, 9 BIV, 6 ALVC	*DSP*, *DSG2*, *PKP2*, *DES*	-Hot-phase represents an uncommon clinical presentation of ACM, which often occurs in pediatric patients and carriers of *DSP* gene mutations. Tissue characterization, family history, and genetic test represent fundamental diagnostic tools for differential diagnosis.
Wang, Europace 2022 [15]	-Aim: to characterize the diagnosis, natural history, and risk for ventricular arrhythmia andheart failure in *DSP* cardiomyopathy	91 patients (49% probands), enrolled in the Johns Hopkins ARVC registry who carry pathogenic or likely pathogenic *DSP* variants	-ALVC forms were common (28%) -Hot-phase episodes in 22% of patients -In univariate regression, myocardial injury was associated with sustained ventricular arrhythmia and HF - LVEF <35% and RV dysfunction were prognostic for sustained ventricular arrhythmia	20 patients (mean age 27.5 years) 14 ALVC	*DSP*	-*DSP* cardiomyopathy affects both ventricles and carries high risk for ventricular arrhythmia and heart failure. -Myocardial injury is associated with worse disease outcomes.
Graziosi, Open Heart 2022 [16]	-Aim: to describe a cohort of patients with ALVC, focusing on the spectrum of the clinical presentations	52 patients (63% M) diagnosed with ALVC retrospectively evaluated	-21 patients (41%) had normal echocardiogram, 13 (25%) a HNDC and 17 (33%) a DCM. -29 (62%) carried a pathogenic/likely pathogenic variant -30 patients (57%) had a previous different diagnosis with a diagnostic delay of 6 years	8 patients (4 M), mean age at symptoms 36 years (min 27, max 60 years)	*DSP*	-ALVC is hidden in different clinical scenarios, with a phenotypic spectrum ranging from normal LV to HNDC and DCM. -Ventricular arrhythmias, chest pain, heart failure and SCD are the main clinical presentations. -Familial screening is essential for the affected relatives’ identification.

ALVC, Arrhythmogenic left ventricular cardiomyopathy; ARVC, Arrhythmogenic right ventricular cardiomyopathy; BIV, Arrhythmogenic biventricular cardiomyopathy; DCM, dilated cardiomyopathy; HNDC, hypokinetic non-dilated cardiomyopathy; LVEF, left ventricular ejection fraction, RVEF, right ventricular ejection fraction; RV, right ventricular; M, males; F, females; SCD, sudden cardiac death; HF, heart failure; LV, left ventricular; *DSP*, Desmoplakin; *PKP2*, Plakofillin-2; *DES*, Desmin; *DSG2*, Desmoiglein-2; and NR: not reported.

**Table 2 biomolecules-12-01324-t002:** Definition of “hot-phase” provided in the nine selected papers.

Reference	Definition
Sen-Chowdhry et al., JACC 2008 [5]	Chest pain and enzyme rise with unobstructed coronary arteries
Lopez Ayala et al., Hearth Rhythm 2015 [10]	Chest pain and enzyme rise with unobstructed coronary arteries. In one patient post-mortem evaluation identified myocardial inflammation
Martins et al., Int J Cardiol 2018 [11]	CMR inflammation criteria
De Witt et al., JACC 2020 [12]	Myocardial inflammation was diagnosed in presence of chest pain with elevated serum troponin with or without ST-segment changes on ECG, in the absence of fever or of infective diseases
Piriou et al., ESC Heart Failure 2020 [13]	Diagnosis of myocardial inflammation was based on the Lake Louise Criteria that were applicable before the end of 2018
Smith et al., Circulation 2020 [14]	Episodic chest pain as a primary symptom independent of arrhythmias, and significant troponin elevation (greater than upper limit of normal as per specific laboratory reference ranges) in the absence of obstructive coronary disease on coronary angiography
Bariani et al., Europace 2021 [8]	Chest pain and myocardial enzyme release in the setting of normal coronary arteries. Eleven patients underwent EMB patients which showed that myocarditis-like features, i.e., Foci of inflammatory infiltration associated with oedema and necrosis of the cardiomyocytes, were found in seven patients
Wang et al., Europace 2022 [15]	Myocardial injury was defined as chest pain, serum cardiac troponin elevation greater than the upper limit of normal, as per local laboratory reference ranges, and the absence of obstructive coronary disease on coronary angiogram
Graziosi et al., Open Heart 2022 [16]	Chest pain: patients requiring hospital admission or outpatient evaluation because of acute or chronic chest pain, respectively

CMR: cardiac magnetic resonance, EMB: endomyocardial biopsy.

**Table 3 biomolecules-12-01324-t003:** Genetic variants reported in ACM patients who showed hot-phase episodes.

Study	N. Subjects	Gene	c.DNA	Amioacid Change	ACMG
Sen Chowdhry et al., JACC 2008 [5]	1	*DSP*	c.3045del	p.Arg1015Serfs*3	Pathogenic
Sen Chowdhry et al., JACC 2008 [5]	1	*DSP*	c.1325C>T	p.Ser442Phe	Likely pathogenic
Lopez-Ajala et al., Heart Rhythm 2015 [10]	1	*DSP*	c.5318del	p.Leu1773Tyrfs*8	Pathogenic
Lopez-Ajala et al., Heart Rhythm 2015 [10]	4	*DSP*	c.1339C>T	p.Gln447*	Pathogenic
Lopez-Ajala et al., Heart Rhythm 2015 [10].	2	*LDB3*	c.1051A>G	p.Thr351Ala	Likely benign
Martins et al., Int J Cardiol 2018 [11]	1	*DSG2*	c.2410del	p.Thr804Leufs*4	Likely pathogenic
Martins et al., Int J Cardiol 2018 [11]	1	*PKP2*	c.2062T>C	p.Ser688Pro	Likely pathogenic
Martins et al., Int J Cardiol 2018 [11]	1	*DSP*	c.8392_8393del	p.Thr2798Trpfs*53	Likely pathogenic
Martins et al., Int J Cardiol 2018 [11]	1	*DSP*	c.1691C>T	p.Thr564Ile	Likely pathogenic
Martins et al., Int J Cardiol 2018 [11]	1	*PKP2*	c.2014-1G>C		Pathogenic
Martins et al., Int J Cardiol 2018 [11]	1	*DSP*	c.4372C>T	p.Arg1458*	Pathogenic
De Witt et al., JACC 2020 [12]	1	*DES*	c.347A>G	p.Asn116Ser	Pathogenic
De Witt et al., JACC 2020 [12]	1	*DSP*	c.1873C>T	p.Gln625*	Pathogenic
		*DSP*	c.6442G>A	p.Ala2148Thr	VUS
De Witt et al., JACC 2020 [12]	1	*PKP2*	c.1162C>T	p.Arg388Trp	Likely pathogenic
		*PKP2*	c.2301del	p.Glu769Lysfs*31	Likely pathogenic
De Witt et al., JACC 2020 [12]	1	*PKP2*	c.2509del	p.Ser837Valfs*94	Pathogenic
De Witt et al., JACC 2020 [12]	1	*DSP*	c.3526del	p.Val1176Phefs*20	Likely pathogenic
		*DSG2*	c.1003A>G	p.Thr335Ala	VUS
De Witt et al., JACC 2020 [12]	1	*DSP*	c.2920del	p.Thr974Leufs*3	Pathogenic
Piriou et al., ESC Heart Failure 2020 [13]	1	*DSP*	c.3925del	p.His1309Thrfs*40	Likely pathogenic
Piriou et al., ESC Heart Failure 2020 [13]	1	*DSP*	c.1856del	p.Tyr619Serfs*17	Likely pathogenic
Piriou et al., ESC Heart Failure 2020 [13]	1	*DSP*	c.1396C>T	p.Leu466Phe	VUS
		*MYBPC3*	c.1153G>A	p.Val385Met	VUS
Piriou et al., ESC Heart Failure 2020 [13]	1	*DSP*	c.2610del	p.Ile870Metfs*19	Likely pathogenic
Piriou et al., ESC Heart Failure 2020 [13]	1	*DSP*	c.3211C>T	p.Gln1071*	Likely pathogenic
Piriou et al., ESC Heart Failure 2020 [13]	1	*DSG2*	c.146G>A	p.Arg49His	Pathogenic
Bariani et al., Europace 2021 [8]	1	*PKP2*	c.2447_2448del	p.Thr816Argfs*10	Likely pathogenic
Bariani et al., Europace 2021 [8]	1	*DSP*	c.7461_7464del	p.Asp2489Metfs*17	Likely pathogenic
Bariani et al., Europace 2021 [8]	1	*DSG2*	c.2032del	p.Gly679Alafs*3	Likely pathogenic
Bariani et al., Europace 2021 [8]	1	*PKP2*	c.84del	p.Ser29Alafs*10	Likely pathogenic
Bariani et al., Europace 2021 [8]	1	*DSP*	c.3889C>T	p.Gln1297*	Likely pathogenic
Bariani et al., Europace 2021 [8]	2	*DSP*	c.897C>G	p.Ser299Arg	Pathogenic
Bariani et al., Europace 2021 [8]	1	*DES*	c.346A>G	p.Asn116Asp	Likely pathogenic
Bariani et al., Europace 2021 [8]	1	*DSP*	c.2821C>T	p.Arg941*	Pathogenic
Bariani et al., Europace 2021 [8]	1	*DSP*	c.3475G>T	p.Glu1159*	Likely pathogenic
Bariani et al., Europace 2021 [8]	1	*DSP*	c.944G>C	p.Arg315Pro	VUS
Bariani et al., Europace 2021 [8]	2	*DSG2*	c.1672C>T	p.Gln558*	Pathogenic
Bariani et al., Europace 2021 [8]	1	*DSP*	c.6323C>A	p.Ser2108*	Likely pathogenic
Bariani et al., Europace 2021 [8]	2	*PKP2*	c.175C>T	p.Gln59*	Pathogenic
Bariani et al., Europace 2021 [8]	1	*PKP2*	c.1027C>T	p.Gln343*	Likely pathogenic
Bariani et al., Europace 2021 [8]	1	*DSP*	c.3889C>T	p.Gln1297*	Likely pathogenic
Graziosi et al., Open Heart 2022 [16]	1	*DSP*	c.6496C>T	p.Arg2166*	Pathogenic
Graziosi et al., Open Heart 2022 [16]	1	*DSP*	c.2611_2614del	p.Asp871Asnfs*17	Pathogenic
Graziosi et al., Open Heart 2022 [16]	1	*DSP*	c.448C>T	p.Arg150*	Pathogenic
Graziosi et al., Open Heart 2022 [16]	1	*DSP*	c.6850C>T	p.Arg2284*	Pathogenic

## Data Availability

The data presented in this study are available on request from the corresponding author.

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
