# Peer review of "Myocarditis-like Episodes in Patients with Arrhythmogenic Cardiomyopathy: A Systematic Review on the So-Called Hot-Phase of the Disease"

_biomolecules, 2022, doi:10.3390/biom12091324_

Round 1
Reviewer 1 Report
The authors present an interesting systematic review on the characteristics of patients with myocarditis like episodes in the context of arrhythmogenic cardiomyopathy. This arising concept challenges traditional perceptions on arrhythmogenic cardiomyopathy, however there is a striking lack of substantial data on the literature. As such, this review is a welcome appraisal of currently available evidence.
The manuscript is well written with only minor grammatical/syntax errors noted such as:
1. Line 41 “myocytes necrosis” should be “myocyte necrosis”
2. Line 155 “Retrieving studies” could be “Study retrieval”
3. Line 168-169 “ and resulted as being 26 yrs±14 yrs”, could be “with the mean age reported being…”
4. Line 189 “pts” should be “patients”
5. Line 191 “ Regarding degree of electrical instability” could be “With regards to (the degree) of electrical instability..”
6. Line 209 “ Authors” does not need to be capitalized. Also on the same line “should” may be changed to “could“ as this is a suggestion.
7. Line 235 “ enhance” should be “enhanced”
8. Line 272-273 “ a genetic test has been proposed that is an effective tool in the differential diagnosis” is not clear. I assume the authors mean “genetic testing has been proposed as an effective tool facilitating the differential diagnosis”.
9. Line 276-278 “ These data confirm the key-role of a genetic test in diagnostic work-up of cardiomyopathies, even if it is important to consider that gene elusive cases are not rare” could be more clearly written.
10. Line 318 “ did not provide unique results”, it is not clear what “unique” means in this sentence.
Other minor remarks are as follows:
1. The authors should make a short note that inclusion of studies enrolling exclusively DSP mutation carriers may have skewed their results towards an overestimation of DSP as the responsible gene for myocarditis like episodes, although indeed clinical practice suggests DSP mutations are the ones most frequently associated with such expression.
2. Since the authors mention CMR as a key modality in assessing ACM, it would be helpful for the reader to include a representative figure on the phenotype found in such patients.
3. In figure 1, the authors mention meeting exclusion criteria/not meeting inclusion criteria. Is this a legend explaining the reasons studies were excluded or did the authors intend to report number of studies? In any case, this is somewhat confusing and could be removed.
4. For the sake of completeness in terms of the clinical relevance of myocarditis like episodes as discussed in this paper, and especially considering this manuscript comes from a leading group in the field of ACM, readers would welcome a short comment on the following:
1. Considering that “hot-phases” likely have an immunological background, is there any data on immunosuppression as a means to reduce recurrences and possible delay disease progression?
2. Since “hot-phases” are associated with increased arrhythmic activity, is there any suggestion that such events are associated with excess rates of SCD compared to ACM patients without clinically overt myocarditis episodes. In other words, is there any data suggesting that myocarditis like episodes could constitute a possible indication for ICD implantation?
Author Response
The authors present an interesting systematic review on the characteristics of patients with myocarditis like episodes in the context of arrhythmogenic cardiomyopathy. This arising concept challenges traditional perceptions on arrhythmogenic cardiomyopathy, however there is a striking lack of substantial data on the literature. As such, this review is a welcome appraisal of currently available evidence.
The manuscript is well written with only minor grammatical/syntax errors noted such as:
- Line 41 “myocytes necrosis” should be “myocyte necrosis”
Thank you. Following reviewer’s suggestion, the word "myocytes" was replaced with "myocyte" (page 2, line 38).
- Line 155 “Retrieving studies” could be “Study retrieval”
Thank you. Following reviewer’s suggestion, "Retrieving studies" was replaced with "Study retrieval" (page 10, line 156).
- Line 168-169 “and resulted as being 26 yrs±14 yrs”, could be “with the mean age reported being…”
Thank you. Following reviewer’s suggestion, changes were made in page 10, lines 170-171.
- Line 189 “pts” should be “patients”
Thank you. The term “pts” was replaced with “patients” throughout the manuscript.
- Line 191 “Regarding degree of electrical instability” could be “With regards to (the degree) of electrical instability”
Thank you. Following reviewer’s suggestion, change were made in page 11, line 203.
- Line 209 “ Authors” does not need to be capitalized. Also on the same line “should” may be changed to “could“ as this is a suggestion.
Thank you. Following reviewer’s suggestion, changes were made in page 15, line 225.
- Line 235 “ enhance” should be “enhanced”
Thank you. Following reviewer’s suggestion, change has been made in page 15, line 257.
- Line 272-273 “a genetic test has been proposed that is an effective tool in the differential diagnosis” is not clear. I assume the authors mean “genetic testing has been proposed as an effective tool facilitating the differential diagnosis”.
We thank the reviewer for the suggestions. The text was modified (page 16, line 296).
- Line 276-278 “ These data confirm the key-role of a genetic test in diagnostic work-up of cardiomyopathies, even if it is important to consider that gene elusive cases are not rare” could be more clearly written.
Thank you for the suggestion. The sentence was modified (page 17, lines 303-306).
- Line 318 “ did not provide unique results”, it is not clear what “unique” means in this sentence.
Thank you for the suggestion. The sentence was modified (page 17, lines 350-353).
Other minor remarks are as follows:
- The authors should make a short note that inclusion of studies enrolling exclusively DSP mutation carriers may have skewed their results towards an overestimation of DSP as the responsible gene for myocarditis like episodes, although indeed clinical practice suggests DSP mutations are the ones most frequently associated with such expression.
Thank you for your suggestion. A sentence discussing the possible selection bias was added (page 16, lines 298-302).
- Since the authors mention CMR as a key modality in assessing ACM, it would be helpful for the reader to include a representative figure on the phenotype found in such patients.
Thanks for the advice. A figure showing the characteristics of cardiac magnetic resonance imaging during two hot phase episodes was added (Fig. 2).
- In figure 1, the authors mention meeting exclusion criteria/not meeting inclusion criteria. Is this a legend explaining the reasons studies were excluded or did the authors intend to report number of studies? In any case, this is somewhat confusing and could be removed.
Thank you for your comment. The PRISMA diagram was modified.
- For the sake of completeness in terms of the clinical relevance of myocarditis like episodes as discussed in this paper, and especially considering this manuscript comes from a leading group in the field of ACM, readers would welcome a short comment on the following:
- Considering that “hot-phases” likely have an immunological background, is there any data on immunosuppression as a means to reduce recurrences and possible delay disease progression?
Immunosuppression in these patients could be a therapeutic option (new ref #38). However, only one case report on this topic is available so far, while no case-control studies exist so far.
- Since “hot-phases” are associated with increased arrhythmic activity, is there any suggestion that such events are associated with excess rates of SCD compared to ACM patients without clinically overt myocarditis episodes. In other words, is there any data suggesting that myocarditis like episodes could constitute a possible indication for ICD implantation?
As stated in the Discussion (page 17, line 313-314) data on degree of electrical instability during the hot-phases are limited and papers are often difficult to compare due to the different selection of patients. Overall, data on arrhythmic burden were available in 54% of our cohort, reporting 18% of patients with SCD or sVT episodes. While in these patients detection of troponin release usually suggests a continuous monitoring in a hospital until the end of the acute episode, data on arrhythmic risk during follow-up are lacking. A sentence which summarizes these concepts was added to the text (page 17, lines 321-324).
Reviewer 2 Report
In the current manuscript, Bariani et al. sought to characterize the presence of myocarditis-like "hot-phase" episodes in the arrhythmogenic cardiomyopathy (ACM) population in the form of a systematic review. The authors performed a systematic search on established databases for publications linking "hot-phase" to ACM (characterised by chest pains accompanied by myocardial enzmes biomarkers, ECG abnormalities), and derived 9 studies matching their criteria. They superficially identify clinical and genetic features in ACM patients associated with "hot-phase". While the topic is in itself interesting, the main clinical question of whether "hot-phase" episodes may identify severity or time progression of ACM patients is not able to be sufficiently addressed in this systematic review.
Major comments:
1) Abstract is too lengthy. Especially intro - a lot of repetition in introduction - can be made concise. Needs a method section e.g. We performed a search on [database1-3] on the keywords and so on... [Intro/Aim]>[Methods]>[Results]>[Conclusion]
2) Line 60: The aim set by authors is rather subjective and unclear to the reader. The aim "to assess the level of knowledge on the "hot-phase" phenomenon in ACM patients" can be misconstrued to suggest a totally different study in the form of a survey in ACM patients on their "level of knowledge" on "hot-phase" phenomenon. It will be more appropriate to suggest that you sought to characterize the clinical features in ACM patients presenting with "hot-phase" episodes.
Also, as the last paragraph of the introduction, you should specify what the current gap in literature exists that your study is hoping to address for clarity to the reader. How is this new knowledge/evidence important clinically?
3) Figure 1 - 1) It is not mentioned in methods what automation tools were used to identify ineligible records, 2) "removed for other reasons" is unacceptable - please specify what these reasons may be either in figure or as a sub-text in figure legend, 3) Reports excluded meeting "exclusion" and "not meeting inclusion" criteria should be identified individually rather as a compound whole.
4) A major criticism of the current manuscript is the lack of comparison with "hot-phase" negative ACM patients population that fails to identify if "hot-phase" are suggestive of a worse outcome or perhaps a prior indicator of ACM (since authors appear to track age of presentation). Certainly, a meta-analysis will be more informative in this context, but without, a more descriptive Results section will be of help to the reader.
Line 166 -103 ACM patients (w/ hot-phase history) - what is the prevalence of hot-phase in the overall ACM population (what's the min-max % prevalence between the 9 studies; any suggestion of risk of bias?)?
Are there possible risk of bias in the studies included leading to the conclusion that patients with hot-phase are usually young/pediatric e.g. for age at presentation e.g. some studies included only <21/ <18 ages?
Line 170-176: Not clear - consider re-wording for clarity. it's confusing between ECGs during symptoms vs basal ECG vs resting ECGs (during hot-phase) and their respective %.
Line 174-176: can authors clarify that percentages are correct i.e. total % > 100% (that they are not mutually exclusive). This should be applied to the entire Results section throughout as it is often confusing whether % in brackets refer to the whole population or a subset of said population (e.g. line 184-185: total % receiving a diagnosis = 101% as compared to 88%.
Table 1 can be improved to include standardized details on study population e.g. study population standardize to highlight sample size (% M), age profile of study population vs. hot-phase population to make comparisons easy to identify to the reader. As it stands, only age- and gender- profile is not identified consistently amongst the 9 studies.
Minor comments:
1) Define all abbreviations at first mention e.g. CPK-MB, CMR, among others including gene names (human genes should be italicized whereas protein not)
2) About a third of refs are >10yrs old (excluding the 9 studies included in sys rev); and the original ref is almost 4 decades old (for the historical perspective I'm guessing) - please check if there are more recent refs on the topic/data and whether still appropriate/relevant in context.
3) Figure 2 - Check for copyright issues with including an image from ref [7].
4) I don't think Ref 9-11 which refers to said databases are actually necessary/adds value - perhaps journal editor can better advise on this.
Author Response
In the current manuscript, Bariani et al. sought to characterize the presence of myocarditis-like "hot-phase" episodes in the arrhythmogenic cardiomyopathy (ACM) population in the form of a systematic review. The authors performed a systematic search on established databases for publications linking "hot-phase" to ACM (characterised by chest pains accompanied by myocardial enzymes biomarkers, ECG abnormalities), and derived 9 studies matching their criteria. They superficially identify clinical and genetic features in ACM patients associated with "hot-phase". While the topic is in itself interesting, the main clinical question of whether "hot-phase" episodes may identify severity or time progression of ACM patients is not able to be sufficiently addressed in this systematic review.
Major comments:
- Abstract is too lengthy. Especially intro - a lot of repetition in introduction - can be made concise. Needs a method section e.g. We performed a search on [database1-3] on the keywords and so on... [Intro/Aim]>[Methods]>[Results]>[Conclusion]
Following the reviewer's suggestions, the abstract was modified (page 1, lines 16-35).
- Line 60: The aim set by authors is rather subjective and unclear to the reader. The aim "to assess the level of knowledge on the "hot-phase" phenomenon in ACM patients" can be misconstrued to suggest a totally different study in the form of a survey in ACM patients on their "level of knowledge" on "hot-phase" phenomenon. It will be more appropriate to suggest that you sought to characterize the clinical features in ACM patients presenting with "hot-phase" episodes.
Thank you for the comment. The sentence was modified as suggested by the reviewer ( page 2, lines 60-61).
- Also, as the last paragraph of the introduction, you should specify what the current gap in literature exists that your study is hoping to address for clarity to the reader. How is this new knowledge/evidence important clinically?
Thank you for the suggestion. Currently, the role of hot-phases in disease progression and arrhythmic risk stratification, remains unclear. This information was added both in the Abstract (page 1, lines 34-35) and in the Introduction (page 2, lines 58-59).
- Figure 1 - 1) It is not mentioned in methods what automation tools were used to identify ineligible records, 2) "removed for other reasons" is unacceptable - please specify what these reasons may be either in figure or as a sub-text in figure legend, 3) Reports excluded meeting "exclusion" and "not meeting inclusion" criteria should be identified individually rather as a compound whole.
We thank the Reviewer for the suggestions. 1) MeSH terms and keywords were combined accordingly on the aforementioned databases. The reference lists of all the included articles were accurately screened in order to identify other pertinent studies. This information was added in the Methods section (page 2, lines 71-73).
2) “Removed for the other reasons” was replaced with “no data of interest” in the PRISMA flow diagram.
3) Some of the manuscripts excluded as “meeting exclusion criteria” and “not meeting inclusion criteria” were reported in the references list and their content was reported in the discussion (ref 6, 7, 17-32)
- A major criticism of the current manuscript is the lack of comparison with "hot-phase" negative ACM patients population that fails to identify if "hot-phase" are suggestive of a worse outcome or perhaps a prior indicator of ACM (since authors appear to track age of presentation). Certainly, a meta-analysis will be more informative in this context, but without, a more descriptive Results section will be of help to the reader.
Thank you for your comment. A sentence was added reporting the main weaknesses of the current literature (page 2, lines 60-61, page 12, lines 243-248). Concerning the major criticism, systematic reviews aim to summarize the results of all relevant studies on a specific topic, thus making the available results more accessible. Comparison with a cohort of ACM patients who do not present hot phase is beyond the scope of this paper, which was instead to collect the current knowledge on the clinical features of the hot phase phenomenon. We agree that a meta-analysis should be informative in this sense.
- Line 166 -103 ACM patients (w/ hot-phase history) - what is the prevalence of hot-phase in the overall ACM population (what's the min-max % prevalence between the 9 studies; any suggestion of risk of bias?)?
The mean observed incidence of hot phase was 12±7% (min 2% - max 22%). The high variability is probably due to variability in patients selection. In some studies, hot phase constitutes the inclusion criterion (Lopez-Ajala [10], Martin et al. [11] and Bariani et al. [8]), but in most of studies inclusion criteria was ACM diagnosis and hot-phases were reported among the other clinical features. This information was added in the text (9, lines 168-169).
- Are there possible risk of bias in the studies included leading to the conclusion that patients with hot-phase are usually young/pediatric e.g. for age at presentation e.g. some studies included only <21/ <18 ages?
Thank you for your comments. We could speculate that the inclusion of studies reporting only pediatric populations could lead to a bias on age at hot phase presentation. However, the two selected cohorts of pediatric patients (ref 11 and 12) represent a small percentage of the overall analyzed population (11/103, 11%). Furthermore, analysis of the 92 pts described in the remaining seven studies showed that in 40% of cases the age at the time of hot phase was below 8 yrs. This information was added in the Discussion chapter (page 16, lines 262-267).
- Line 170-176: Not clear - consider re-wording for clarity. it's confusing between ECGs during symptoms vs basal ECG vs resting ECGs (during hot-phase) and their respective %.
Thanks for the suggestion. The sentence has been modified (page 10, lines 172-177).
- Line 174-176: can authors clarify that percentages are correct i.e. total % > 100% (that they are not mutually exclusive). This should be applied to the entire Results section throughout as it is often confusing whether % in brackets refer to the whole population or a subset of said population (e.g. line 184-185: total % receiving a diagnosis = 101% as compared to 88%.
Thank you for your comment. The percentages have been checked throughout the results paragraph and when unclear they were modified (page 10, lines 172-177 and page 11 lines 197-202 ).
- Table 1 can be improved to include standardized details on study population e.g. study population standardize to highlight sample size (% M), age profile of study population vs. hot-phase population to make comparisons easy to identify to the reader. As it stands, only age- and gender- profile is not identified consistently amongst the 9 studies.
Thank you for your comment. Unfortunately, not all papers reported in detail the clinical, instrumental and genetic features of patients showing hot phase episodes. In Table 1 we tried to report all available data.
Minor comments:
1) Define all abbreviations at first mention e.g. CPK-MB, CMR, among others including gene names (human genes should be italicized whereas protein not)
Thank you. Corrections were done throughout text.
2) About a third of refs are >10yrs old (excluding the 9 studies included in sys rev); and the original ref is almost 4 decades old (for the historical perspective I'm guessing) - please check if there are more recent refs on the topic/data and whether still appropriate/relevant in context
As the Reviewer correctly points out, some of the references were reported to provide an historical perspective to the reader. In the revised version of the manuscript, new references were added (ref. # 3, #33, #38)
3) Figure 2 - Check for copyright issues with including an image from ref [7].
Permission for reproduction was obtained from the publisher.
4) I don't think Ref 9-11 which refers to said databases are actually necessary/adds value - perhaps journal editor can better advise on this.
Following Reviewer's suggestion, references 9-11 were removed.
Reviewer 3 Report
I have read the review article 'Myocaditis-like episodes in patients with Arrhythmogenic Cardiomyopathy': the so-called hot-phase of the disease. A systematic review, submitted by Riccardo Bariani.
The manuscript is well written and the topic is highly intersting to a broad readership. However, the manuscript needs several changes which will be explained in the following sections:
1.) Please define the term 'hot-phase' with clear criteria in the introduction.
2.) Please explain the genetic background of ACM or add citation of relevant other reviews describing the genetic background within the introduction (e.g. Insights Into Genetics and Pathophysiology of Arrhythmogenic Cardiomyopathy, Gerull B and Brodehl A 2021, Current Heart Failure Reports).
3.) Please specifiy how myocarditis was diagnosed and defined?
4.) Please explain all abbreviations (LGE, ...).
5.) Please add a table with the exact mutations of the patients?
6.) How were mutations classified (VUS, likely pathogenic, pathogenic)?
7.) Please cite also the study of Brodehl A 'Transgenic mice overexpressing desmocollin-2 (DSC2) develop cardimyopathy associated with myocardial inflammation and fibrotic remodeling.' PLOSone 2019 in Line 304 to prevent selective (self-citation).
In summary, a suggest a substantial revision of this interesting and well-written paper! Good luck with the revision!
Author Response
I have read the review article 'Myocaditis-like episodes in patients with Arrhythmogenic Cardiomyopathy': the so-called hot-phase of the disease. A systematic review, submitted by Riccardo Bariani. The manuscript is well written and the topic is highly intersting to a broad readership. However, the manuscript needs several changes which will be explained in the following sections:
- Please define the term 'hot-phase' with clear criteria in the introduction.
Thank you for the comment. The introduction was modified in order to clarify the definition of "hot phase" (page 2, lines 54-56).
- Please explain the genetic background of ACM or add citation of relevant other reviews describing the genetic background within the introduction (e.g. Insights Into Genetics and Pathophysiology of Arrhythmogenic Cardiomyopathy, Gerull B and Brodehl A 2021, Current Heart Failure Reports).
Thank you for the suggestion. The introduction was modified by adding a comment on genetic background and the reference suggested by the reviewer (page 2, lines 40-42 and ref, #3).
- Please specify how myocarditis was diagnosed and defined?
Thank you for your comment. We added a table (Table 2) that summarizes the definitions of the hot phase phenomenon provided in the different studies.
- Please explain all abbreviations (LGE, ...).
Thank you. Corrections were made throughout the text.
- Please add a table with the exact mutations of the patients? How were mutations classified (VUS, likely pathogenic, pathogenic)?
Following the Reviewer's suggestion, we added a new table (Table n. 3) reporting genetic variants reported in the 9 analyzed studies together with variant classification. It should be noted that it was possible to achieve detailed information on genetic variants only in 49 out 85 pts (58%) with positive genetic test. This information was added in the Result chapter (page 11, lines 212-214)
- Please cite also the study of Brodehl A 'Transgenic mice overexpressing desmocollin-2 (DSC2) develop cardimyopathy associated with myocardial inflammation and fibrotic remodeling.' PLOSone 2019 in Line 304 to prevent selective (self-citation).
As suggested by the reviewer, a new reference was added (ref. 33), page 17, line 335.
Round 2
Reviewer 2 Report
The authors have done a good job in addressing my comments and improvements to the manuscript. Only thing is the ECG traces in Figure 3 (old figure 2) is now missing in the latest version.
I have no further suggestions.
Reviewer 3 Report
Congratulations! The authors have addresed my concerns in a sufficient way.